# Study on the Coexistence of Offshore Wind Farms and Cage Culture

Hsing-Yu Wang [1], Hui-Ming Fang [2] and Yun-Chih Chiang [3,*]

1    Department of Shipping Technology, College of Maritime, National Kaohsiung University of Science and Technology, Kaohsiung 80543, Taiwan; hywang05@nkust.edu.tw
2    Bachelor Degree Program in Ocean Engineering and Technology, National Taiwan Ocean University, Keelung 20224, Taiwan; hmfang@email.ntou.edu.tw
3    Center for General Education, Tzu Chi University, Hualien 97004, Taiwan
*    Correspondence: ycchiang@mail.tcu.edu.tw; Tel.: +886-3-856-5301 (ext. 5517)

**Abstract:** In this study, a hydrodynamic model was used that includes the effects of wave–current interactions to simulate the wave and current patterns before and after offshore wind turbine installation in western Taiwan. By simulating the waves and currents after the offshore wind turbine was established, the waves and currents caused by the wind turbine were seen to have a limited range of influence, which is probably within an area about four to five times the size of the diameter (12–15 m) of the foundation structure. Overall, the analysis of the simulation results of the wave and current patterns after the offshore wind turbines were established shows that the underwater foundation only affected the local area near the pile structure. The wind farm (code E) of the research case can be equipped with about 720 cage cultures; if this is extended to other wind farms in the western sea area, it should be possible to produce economic-scale farming operations such as offshore wind power and fisheries. However, this study did not consider the future operation of the entire offshore wind farm. If the operation and maintenance of offshore wind farms are not affected, and if the consent of the developer is obtained, it should be possible to use this method to provide economically large-scale farming areas as a mutually beneficial method for offshore wind power generation and fisheries.

**Keywords:** hydrodynamic model; wind farm; wind turbine; cage culture

## 1. Introduction

Taiwan is a nation surrounded by ocean and has innate conditions for the development of marine applications. The development and utilization of the waters around Taiwan are mostly based on fisheries and shipping. In recent years, with the needs of economic development, the improvement of marine clean energy technology, the rise of marine recreational activities, the development of new type of coasts and oceans, and the increase in marine environmental awareness highlight the problems of space use. Based on the issues of global warming and clean energy demand in recent years, under the pressure of abolishing nuclear energy and reducing carbon emissions, offshore wind power has become one of the options for Taiwan's energy transformation. The offshore wind field in the Taiwan Strait has excellent resources. For example, ref. [1] uses the observation data of average wind speed to show that 13 of the best observation sites in the world are located on the western coast of Taiwan. According to the selection results of the offshore wind power planning site selection [2], the Changhua sea area in western Taiwan has the capacity of offshore wind power installations as high as 62.6%, ranking first in Taiwan's offshore wind power generation capacity distribution area. However, this important area for the development of offshore wind power in the future is also one of the most prosperous areas for offshore fishing operations in western Taiwan. In highly overlapping economic sea

areas, the development of offshore wind power may impact the fishing and ecological environment in the same sea area.

Referring to the problems encountered in the development of offshore wind power in the UK, if an area with very good wind resources is also a resource-rich fishing ground, there will be serious conflicts between the development of wind farms and the livelihoods of local fishermen [3–5]. The potential wind farms that have been announced in Taiwan have a direct impact on the livelihoods of fishermen in the western sea area because of the concentrated location. As the development of offshore wind farms is not yet in place, certain employment opportunities can be created in the future, but there is uncertainty regarding the inclusion of the most affected fishermen. Thus, for fishermen who are unwilling or unable to change their way of living, this issue still needs careful consideration [6–8]. During the development of offshore wind power generation, given the consideration of safety factors during wind farm construction, fishing boats are not allowed to operate in the wind farm construction area, but later stages can allow fishing boats to operate. On the premise of not affecting the maintenance and operation of the wind farm and not violating the wishes of the developer, we consider whether there is opportunity for mutual symbiosis between offshore wind power and fisheries within the scope of the offshore wind farm. Whether the space can provide the necessary factors on an economic scale, with sustainable management costs, and natural disaster response efficiency are questions that can be considered [5,6,9–11]. According to the statistics of the Food and Agriculture Organization of the United Nations (FAO) Fish Stat J database [12], the output value and output of farming fisheries are increasing each year. With the depletion of offshore fisheries resources and the limitation of offshore fisheries due to fishery rights, if offshore wind farms and fisheries can be mutually beneficial and symbiotic as a development opportunity, is it possible to carry out offshore cage culture within the offshore wind farm. Through such development, the goal of co-prosperity of offshore wind power and fisheries can be achieved.

In order to avoid influence of the interaction of the wind turbine's wake, the layout of the generator set is generally to use a distance of up to 4 times the parallel wind direction and a distance of 10 times the vertical wind direction as the interval. This arrangement is mainly to avoid the kinetic energy loss caused by the downstream wind turbines [13–15]. In addition, referring to the relevant environmental protection specifications of the Environmental Impact Assessment Inquiry System [16] for the development of offshore wind farms in the sea area around Taiwan, with the future development of large-scale wind turbines, the offshore wind turbines planned by Taiwan's offshore wind power developers re separated by about 1 km. We consider if it is possible to establish a cage culture system to assist fishermen to transform their livelihoods into sustainable fisheries and to transform offshore wind power and traditional fisheries into cooperative developments in the area within 1 square kilometer. The development utilized by the comprehensive planning of ocean space was thought to be in the long-term public interest for the use of the sea area of the offshore wind power [17,18]. In order to consider the comprehensive evaluation of the use of space in the sea area, this study used the hydrodynamic model under the influence of wave–current interaction to simulate the wave–current pattern around the structure. By simulating the range of waves and current changes that may be affected by offshore wind turbines in order to develop the concept of multiple resource utilization in cage culture, it is hoped that this study can contribute to the prosperity of offshore wind power development and fisheries.

## 2. Methodology

This study used a hydrodynamic model that included wave and current characteristics for calculation. The numerical model first analyzed the overall coastal dynamic characteristics, and corrected some numerical calculation parameters and input/output data formats and contents according to different regions that may be affected, and then analyzed the impact of offshore wind turbines on the wave–current pattern [19–21].

*2.1. Wave Model*

In this study, calculations of wave patterns were carried out using the mild slope equation of the current effect [22]:

$$\frac{D^2\varphi}{Dt^2} + \left(\nabla\cdot\vec{U}\right)\frac{D\varphi}{Dt} - \nabla\cdot(CC_g\nabla\varphi) + \left(\sigma^2 - k^2CC_g\right)\varphi = 0 \tag{1}$$

where $\vec{U}$ is the ambient current, $\nabla$ is the horizontal gradient operator, $\varphi$ is the two-dimensional velocity potential, $k$ is the wave number, $C$ and $C_g$ are the phase and group speed of the waves, and $\sigma$ is the dispersion relation given by $\sigma^2 = gk\tanh kh$. Under the assumption of the irrotational field, single-frequency linear surface waves, the potential energy of a wave can be expressed as follows:

$$\varphi\left(\vec{x},\vec{y},z,t\right) = f(z,h)\varphi\left(\vec{x},\vec{y},t\right) \tag{2}$$

where $f(z,h) = \frac{\cosh[k(h+z)]}{\cosh kh}$. In a single periodic harmonic motion, Equation (2) can be rewritten as follows:

$$\varphi\left(\vec{x},\vec{y},t\right) = \mathrm{Re}\left\{ae^{is}e^{i\omega t}\right\} \tag{3}$$

The following expression can be obtained by substituting Equation (3) into Equation (1) for Equation (4)'s real part and Equation (5)'s imaginary part:

$$\frac{1}{aCC_g}\left\{\left(\vec{U}\cdot\nabla a\right)\left[\left(\vec{U}\cdot\nabla\right) + \left(\nabla\cdot\vec{U}\right)\right]\right\} - \frac{1}{a}\left[\nabla^2 a + \frac{1}{CC_g}\left(\nabla CC_g\cdot\nabla a\right)\right] - k^2 + |\nabla s|^2 = 0 \tag{4}$$

$$\nabla\cdot\left[a^2\sigma\left(U + C_g\right)\right] = 0 \tag{5}$$

Equations (4) and (5) are the equations of motion for wave interaction before breaking waves. When the current velocity $\vec{U}$ is known, it solves the system of linear equations in two unknown parabolic simultaneous equations and obtains the amplitude $a(x,y)$ and the wave number $|\nabla s|$. When $\vec{U} = 0$, Equations (4) and (5) become Equations (6) and (7):

$$\frac{1}{a}\left\{\frac{\partial^2 a}{\partial x^2} + \frac{\partial^2 a}{\partial y^2} + \frac{1}{CC_g}\left[\nabla a\cdot\nabla\left(CC_g\right)\right]\right\} + k^2 - |\nabla s| = 0 \tag{6}$$

$$\nabla\cdot\left[a^2 CC_g\nabla s\right] = 0 \tag{7}$$

In addition, energy was dissipated in the surf zone, and the energy expression of Equation (5) must be modified. Based on the energy flux theory, ignoring the effect of bottom friction [23]:

$$\frac{d\left(EC_g\right)}{dx} = -\varepsilon, \; \varepsilon = \frac{1}{2}\rho V_e(kH_B)^2, \; V_e = V_{eB}\left(\frac{H_B/2 - c\prime h_B}{\gamma\prime h_B}\right)^m$$
$$, V_{eB} = \frac{5S_B g}{8k_B\rho}\frac{1}{\sqrt{1-C_0}}, \; S_B == \frac{\tan\beta}{1+\frac{3r^2}{2}} \tag{8}$$

where $c\prime$ is the ratio of the radiation to the water depth of the recovery zone. According to [23], $c\prime = 0.17$ when the wave recovery zone is not obvious in a gentle slope.

In the area of wave–current interaction, the energy dissipated by the nearshore current within the surf zone is small and negligible, so the energy amplitude expression according to Equation (8) can be expressed as follows:

$$\nabla\cdot\left[\frac{E}{\sigma}\left(\vec{U} + C_g\right)\right] = -\frac{5}{16}\frac{\rho g^2 k_B}{\sigma^2}\frac{\tan\beta}{1+\frac{3r^2}{2}}\frac{1}{\sqrt{1-C_0}}\sqrt{\frac{H_B/2 - c\prime h_B}{r\prime h_B}}(H_B)^2 \tag{9}$$

With Equation (6), the energy in the surf zone has been expressed, and Equation (9) has been modified as follows in Equation (10):

$$\nabla \cdot \left[ a^2 \sigma \left( \vec{U} + C_g \right) \right] = \nabla \cdot \left[ \frac{2g}{\rho} \frac{E}{\sigma} \left( \vec{U} + C_g \right) \right] = -\frac{5}{8} \frac{g^2 k_B}{\sigma} \frac{\tan \beta}{1 + \frac{3r/2}{2}} \frac{1}{\sqrt{1 - \frac{c/}{r/}}} \sqrt{\frac{H_B/2 - c/h_B}{r/h_B}} (H_B)^2 \quad (10)$$

In Equations (8)–(10), the subscript *B* indicates the value at the surf zone. As the phase function of $\phi$ is $x\left( \vec{x}, t \right) = s\left( \vec{x} \right) - \omega t$, the wave number obtained from the deformed mild slope equation can be expressed as follows:

$$\vec{k} = \nabla x = \nabla s \quad (11)$$

To obtain $|\nabla s|$ from Equations (4), (5), or (10), it is necessary to know the direction of the wave. There are only two equations to solve $a$, $|\nabla s|$, $\theta$. The linearity of the wave phase function gradient is assumed to be irrotational by Equation (12), and the convergence conditions of wave model is given by Equation (13):

$$\begin{aligned}
&\nabla \times (\nabla s) = 0 \\
&\nabla s = |\nabla s| \cos \theta \, \vec{i} + |\nabla s| \sin \theta \, \vec{j} \\
&\frac{\partial}{\partial x} (|\nabla s| \sin \theta) - \frac{\partial}{\partial y} (|\nabla s| \cos \theta) = 0
\end{aligned} \quad (12)$$

$$\begin{aligned}
&|H_{now} - H_{old}| \leq \varepsilon_H (H_{now}) \,, \quad \varepsilon_H = 0.001 \\
&|H_{1now} - H_{1old}| \leq \varepsilon_k (H_{1now}) \,, \quad \varepsilon_k = 0.001 \\
&|H_{2now} - H_{2old}| \leq \varepsilon_k (H_{2now}) \,, \quad \varepsilon_k = 0.001
\end{aligned} \quad (13)$$

### 2.2. Hydrodynamic Model

In this study, the tidal effect was added to the hydrodynamic model. In addition to the tide, which is regarded as a long wave, the Coriolis force effect of the earth's rotation was also considered. The governing equations of the hydrodynamic model are as follows:

Continuity equation:

$$\frac{\partial \eta}{\partial t} + \frac{\partial}{\partial x} [U(h + \eta)] + \frac{\partial}{\partial y} [V(h + \eta)] = 0 \quad (14)$$

Momentum equations:

$$\begin{aligned}
&\frac{\partial U}{\partial t} + U \frac{\partial U}{\partial x} + V \frac{\partial U}{\partial y} = fV - g \frac{\partial \eta}{\partial x} + \frac{1}{\rho} \left( \frac{\partial \tau_{xx}}{\partial x} + \frac{\partial \tau_{yx}}{\partial y} \right) + \frac{1}{\rho(h + \eta)} \left( \tau_{sx} - \tau_{bx} \right) \\
&\qquad - \frac{1}{\rho(h + \eta)} \left( \frac{\partial S_{xx}}{\partial x} + \frac{\partial S_{yx}}{\partial y} \right)
\end{aligned} \quad (15)$$

$$\begin{aligned}
&\frac{\partial V}{\partial t} + U \frac{\partial V}{\partial x} + V \frac{\partial V}{\partial y} = -fU - g \frac{\partial \eta}{\partial y} + \frac{1}{\rho} \left( \frac{\partial \tau_{xy}}{\partial x} + \frac{\partial \tau_{yy}}{\partial y} \right) + \frac{1}{\rho(h + \eta)} \left( \tau_{sy} - \tau_{by} \right) \\
&\qquad - \frac{1}{\rho(h + \eta)} \left( \frac{\partial S_{xy}}{\partial x} + \frac{\partial S_{yy}}{\partial y} \right)
\end{aligned} \quad (16)$$

where $\eta$ is the water surface elevation, $h$ is the static water depth, $U$ and $V$ are the average velocity components of the water depth in the fixed coordinates of the $x$ and $y$ axes:

$$U = \frac{1}{(h + \eta)} \int_{-h}^{\eta} u \, dz \,, \quad V = \frac{1}{(h + \eta)} \int_{-h}^{\eta} v \, dz \quad (17)$$

Shear stress $\tau_{xx}$, $\tau_{xy}$, $\tau_{yx}$, $\tau_{yy}$ includes viscous stress caused by fluid viscosity and Reynolds stress caused by turbulent effects. As the value of viscous stress compared with

Reynolds stress is very small, viscous stress is ignored generally, and only Reynolds stress is considered to represent the momentum exchange between fluids:

$$\tau_{xx} = \rho E_v \frac{\partial U}{\partial x} \quad , \quad \tau_{xy} = \rho E_v \frac{\partial U}{\partial y} \quad , \quad \tau_{yx} = \rho E_v \frac{\partial V}{\partial x} \quad , \quad \tau_{yy} = \rho E_v \frac{\partial V}{\partial y} \tag{18}$$

The vortex viscosity coefficient $E_v$ is obtained from the semi-empirical formula of the Prandtl mixing length theory [24]:

$$E_v = \frac{k_v \sqrt{g}(d + h)\sqrt{U^2 + V^2}}{6 C_c} \tag{19}$$

The sea surface wind shear components $\tau_{sx}$ and $\tau_{sy}$ are the components of the sea surface wind shear in the $x$ and $y$ directions; refer to [25]:

$$\tau_{sx} = \rho k_w W^2 \cos a \quad ; \quad \tau_{sy} = \rho k_w W^2 \sin a \tag{20}$$

$$k_w = \begin{cases} 1.2 \times 10^{-5} & , \quad W \le W_c \\ 1.2 \times 10^{-6} + 2.25 \times 10^{-6}\left[1 - \frac{W_c}{W}\right]^2 & , \quad W > W_c \end{cases} \tag{21}$$

The bottom friction stress $\tau_{bx}$ and $\tau_{by}$ are the components in $x$ and $y$ directions [26]:

$$\tau_{bx} = \rho E_r U \sqrt{U^2 + V^2} \quad ; \quad \tau_{by} = \rho E_r V \sqrt{U^2 + V^2} \tag{22}$$

where the coefficient of the bottom friction $F_r = g/C_c^2$.

The radiation stress $S_{xx}$, $S_{xy}$, $S_{yx}$, $S_{yy}$, which is expressed by the linear wave theory as follows [27] is the main factor causing longshore currents:

$$\begin{bmatrix} S_{xx} & S_{xy} \\ S_{yx} & S_{yy} \end{bmatrix} = \overline{E} \begin{bmatrix} n\left(1 + \cos^2 \theta\right) - \frac{1}{2} & \left(\frac{n}{2}\right)\sin(2\theta) \\ \left(\frac{n}{2}\right)\sin(2\theta) & n\left(1 + \sin^2 \theta\right) - \frac{1}{2} \end{bmatrix} \tag{23}$$

where $\overline{E}$ is the total wave energy per unit time and area of section. Under the Airy wave theory:

$$\overline{E} = \frac{\rho g H^2}{8} \tag{24}$$

The boundary conditions of the hydrodynamic model are shown in Figure 1. The water level change includes the level of rise and fall caused by waves and tides.

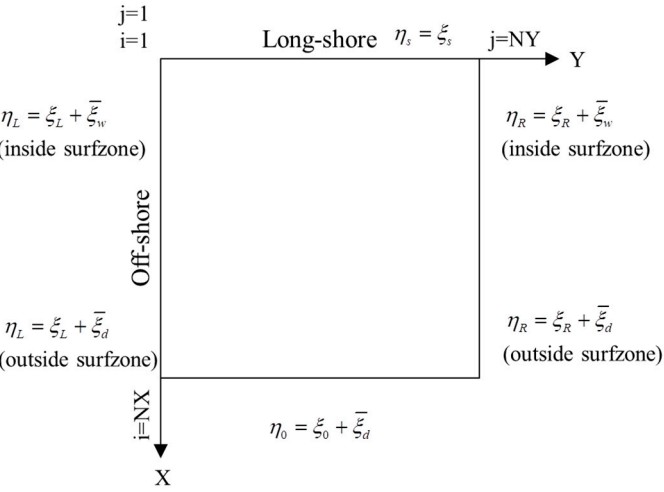

**Figure 1.** Hydrodynamic model boundary conditions.

Tide is a sine function changing from right to left, $T_L$ is the time difference when the tide reaches the left and right boundaries, and $T_{start}$ is the phase difference:

Left boundary:

$$\eta_L = A_L^t \sin\left[\frac{2\pi}{T_t}(t + T_t + T_{start})\right] , \quad T_t = \frac{L_y}{\sqrt{gh_{Max}}} \tag{25}$$

Right boundary:

$$\eta_R = A_R^t \sin\left[\frac{2\pi}{T_t}(t + T_{start})\right] \tag{26}$$

Offshore boundary:

$$\eta_0 = \left[A_R^t + (A_L^t - A_R^t)\left(\frac{N_y - j}{N_y - 1}\right)\right] \sin\left\{\frac{2\pi}{T_t}\left[t + T_t\left(\frac{N_y - j}{N_y - 1}\right) + T_{start}\right]\right\} \tag{27}$$

The water level $\overline{\overline{\zeta}}$ caused by waves is based on [28], ignoring the reflection effect. Equation (28) is the water level descent outside the surf zone, and Equation (29) is the water level uplift in the surf zone:

$$\overline{\zeta_d} = -\frac{H^2}{8}\frac{k}{\sinh(2kh)}(\cos\theta)^{2/3} \tag{28}$$

$$\frac{d\overline{\zeta_u}}{dx} = -K\frac{dh}{dx} , \quad K = \frac{1}{1 + (8/3\gamma^2)} \tag{29}$$

The left boundary of the velocity is shown in Equation (30), the right boundary is shown in Equation (31), the offshore boundary is shown in Equation (32), and the longshore boundary is shown in Equation (33).

$$U_{j=1} = U_{j=2} , \quad \left(\frac{\partial V}{\partial y}\right)_{j=1} = 0 \tag{30}$$

$$U_{j=NY} = U_{j=NY-1} , \quad \left(\frac{\partial V}{\partial y}\right)_{j=NY} = 0 \tag{31}$$

$$U_{j=NX} = U_{j=NX-1} , \quad \left(\frac{\partial U}{\partial x}\right)_{j=NX} = 0 \tag{32}$$

$$U = 0 , \quad V = 0 \tag{33}$$

The stability of the hydrodynamics calculation must satisfy $\Delta t \leq 2\Delta s / \sqrt{gh_{Max}}$, where $\Delta s$ is the grid size. The maximum value of two adjacent time steps is less than the allowable error with Equation (34), and then the calculation of the next time step can be performed.

$$\begin{aligned} Max\left(\eta_{ij}^{k+1} - \eta_{ij}^k\right) &\leq \varepsilon_\eta \eta_{ij}^k , \quad \varepsilon_\eta = 0.0001 \\ Max\left(U_{ij}^{k+1} - U_{ij}^k\right) &\leq \varepsilon_U U_{ij}^k , \quad \varepsilon_U = 0.0001 \\ Max\left(V_{ij}^{k+1} - V_{ij}^k\right) &\leq \varepsilon_V V_{ij}^k , \quad \varepsilon_V = 0.0001 \end{aligned} \tag{34}$$

### 2.3. Hydrodynamic Characteristics at the Coast

The numerical process is to first calculate the distribution of the wave pattern, and then use the radiation stress and the change mechanism of the tide level to analyze the mixed current pattern including near-shore currents. According to the principle of zoning for offshore wind farms in Taiwan, 36 potential areas (different colors) can be planned from north to south, as shown in Figure 2 [29]. It can be seen from Figure 2 that most of the potential areas are concentrated in western Taiwan. Since most offshore wind turbines are set up at a depth of 30 m along the coast, and the offshore wind farm will form a

certain area of obstacles, this may affect fishing operations and will have a direct impact on the livelihood of fishermen. Therefore, this study will establish numerical topography for the area closer to the shore. The water depth data needed to establish the numerical topography mainly collect the water depth as measured by the 4th River Management Office (Water Resources Agency) and the large-scale water depth data provided by Ocean Data Bank (Ministry of Science and Technology). This study will establish numerical topography for the area closer to the shore. The grid size for the wave field simulation must meet the 1/8 wavelength, but most developers of offshore wind power generation in Taiwan currently propose to use the jacket-type offshore wind turbine support structure, such as that shown in Figure 3 [30]. As the diameter of the structure using the jacket-type is not large, considering the calculation efficiency and error, the spatial grid sizes of $\Delta x = \Delta y = 0.5$m were used in wave and hydrodynamic model, and the time step interval of $\Delta t = 1$s was used for the model. The calculation conditions of the numerical model were mainly obtained by the observation station and referring to the field observation data of The Feasibility Study of Offshore Wind Power in Changhua: Phase One [31]. Table 1 lists the configuration settings of the hydrodynamic model. Additionally, the numerical topography created from these data are shown in Figure 4, and the water depth distribution was between $-10$ and $-50$ m. Figure 5 compares the model simulation results with the current measurement. The calculated current velocity, current direction, and changes in water level were all highly consistent with the measured data. The currents simulated by the model generally flowed parallel to the coastline. Specifically, the current flowed from the southwest to the northeast during flood tides and from the northeast to the southwest during ebb tides. The results, in accordance with the actual situation, indicate that the model successfully reproduced the characteristics of the currents in the sea area. After the boundary conditions and related parameters adopted by the hydrodynamic model were determined, the model was employed to calculate the hydrodynamic characteristics before and after the offshore wind turbine was established.

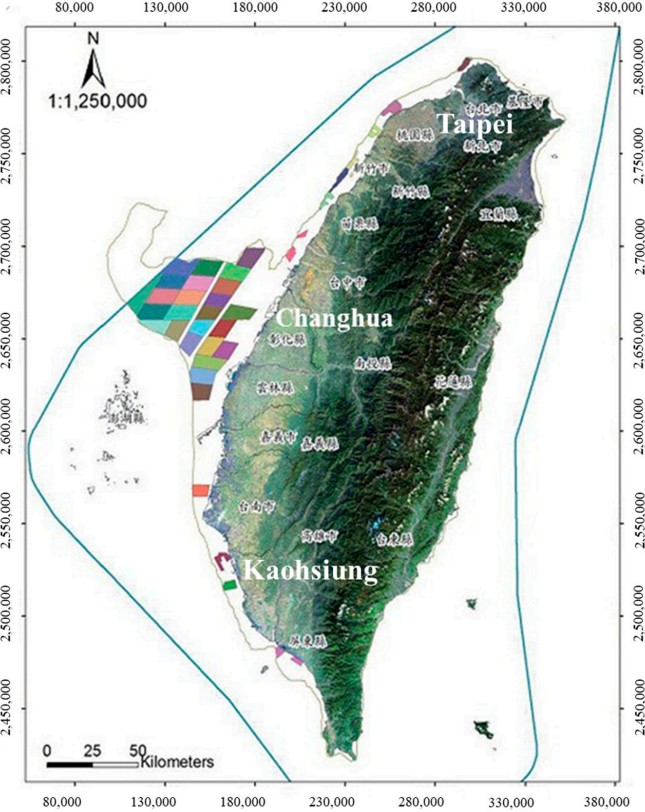

**Figure 2.** Schematic diagram of Taiwan's offshore wind power generation area. Source: Ministry of Economic Affairs, Taiwan (2018).

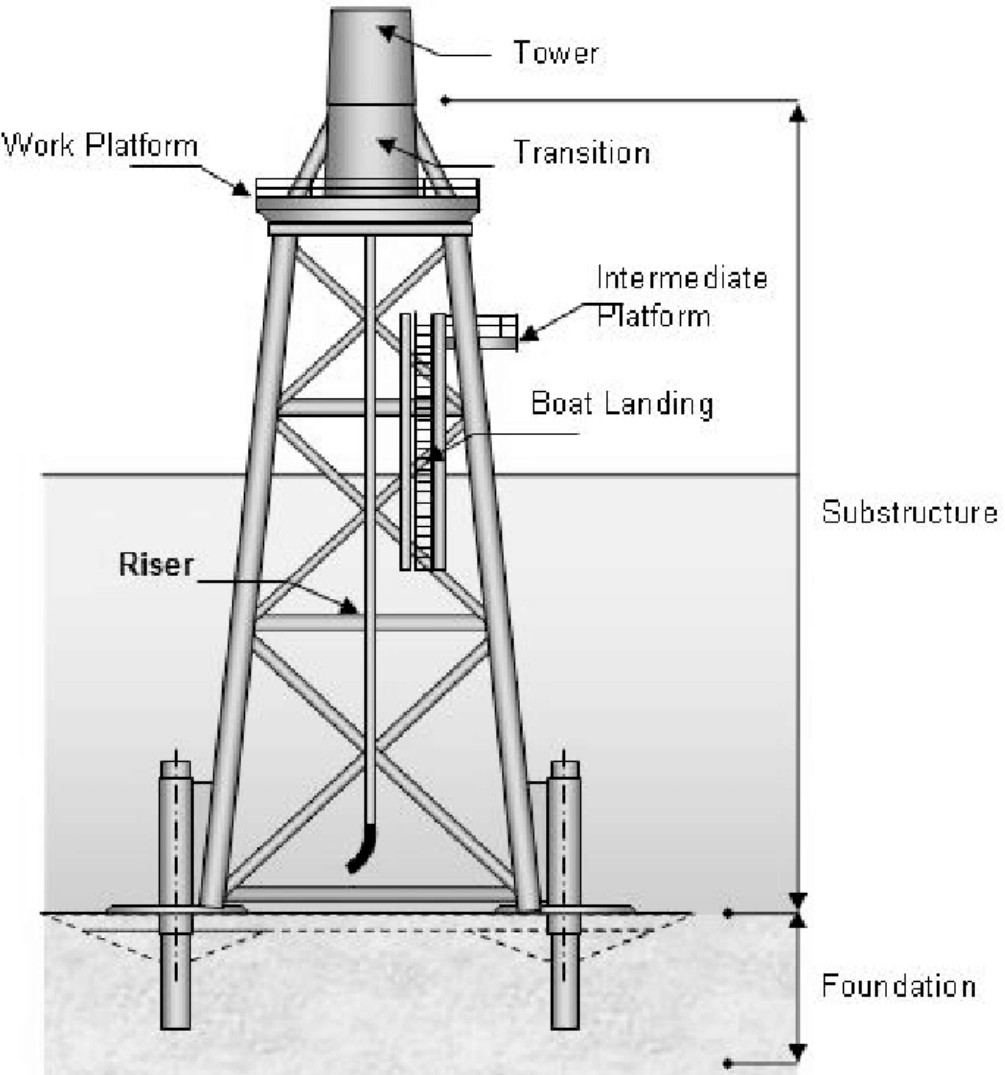

**Figure 3.** The jacket-type offshore wind turbine support structure [30].

**Table 1.** Calculation conditions for wave and hydrodynamic model calibration.

| Item | Model Setup |
|---|---|
| Area | 50 km × 35 km |
| Grid size | 0.5 m × 0.5 m |
| Number of grid points | 100,000 × 70,000 |
| Coordinate of the origin (TWD97) | 168,110, 2621,287 |
| Angle of deviation (counterclockwise from the north) | 60° |
| Time step size | 1.0 s |
| Wave condition | SSW direction; H = 1.2 m; T = 6.5 s |
| [31] | NNW direction; H = 3.0 m; T = 7.5 s |

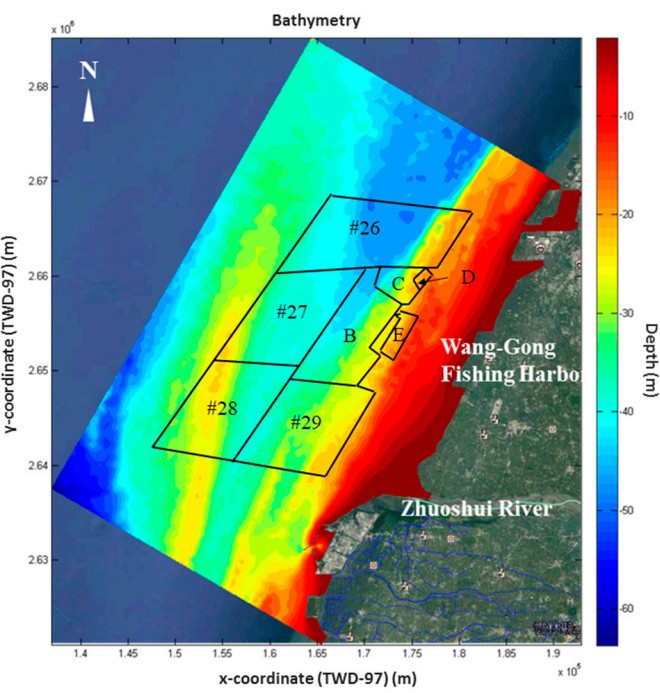

**Figure 4.** Establishment of numerical topography in Changhua sea area.

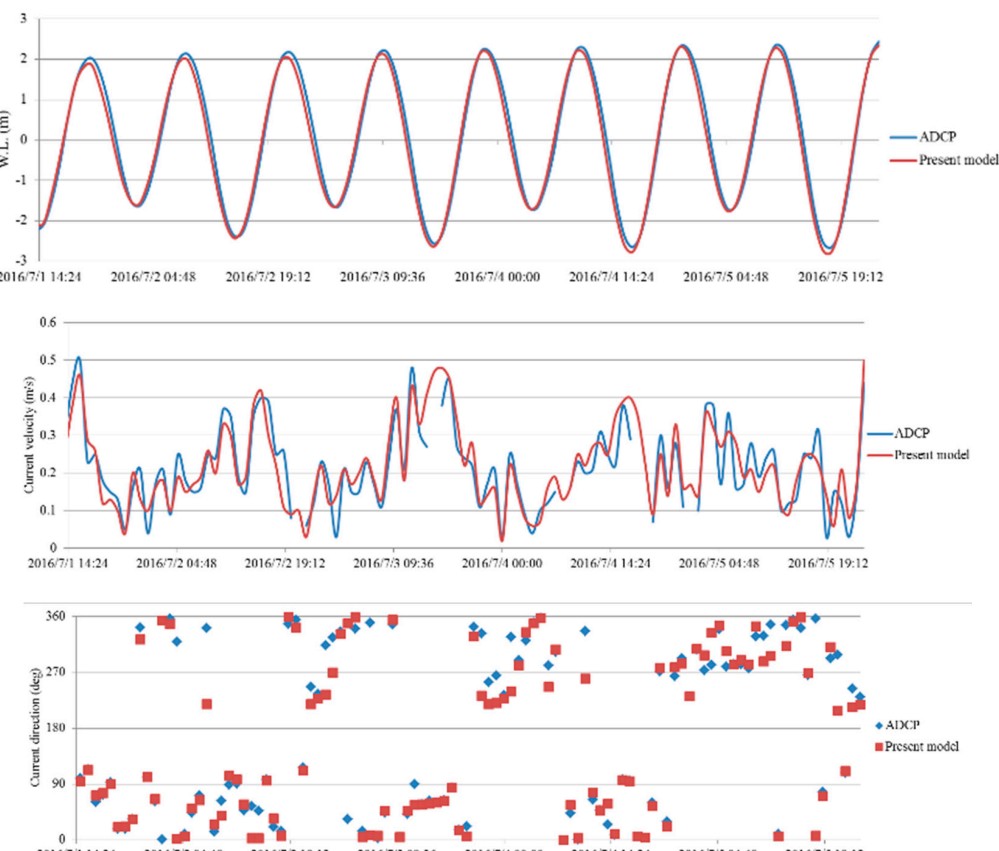

**Figure 5.** Comparison between the numerical model and the measured data. Blue line, measurement data; Red line, simulation results.

This study referred to the technical content of The Feasibility Study of Offshore Wind Power in Changhua: Phase One [31] to set the mode input conditions. In summer, the wave height was 1.2 m, the period was 6.5 s, and the wave direction was south-southwest (SSW);

in winter, the wave height was 3.0 m, the period was 7.5 s, and the wave direction was north–northwest (NNW). In order to make the results clearer and more convenient, this study will narrow down the calculation results. The simulation results of the summer and winter wave and current patterns are shown in Figures 6a and 7a. The summer monsoon wave was attenuated sharply at a depth of about 0–2 m near the shore, which means that most of the wave energy was released in this area; the winter monsoon wave was also attenuated sharply at a water depth of 0–2 m. The diagrams of the near-shore current simulated by Figure 6b,c and Figure 7b,c show that the phenomenon of near-shore currents generated by waves before and after breaking waves was quite obvious. During the wave impact of the summer monsoon, the direction of the current between a 0 and 2 m water depth was approximately from the southwest to the northeast, and a vortex also occurred in some areas between a −1 and −3 m water depth nearing the protruding coastal structures. Additionally, during the winter monsoon, a large near-shore current velocity was found between the depth of −1 and −5 m, and the overall transmission direction was roughly southwest–northeast. The average velocity of the overall offshore wind farm area was between 0.56 and 0.58 m/s, and the maximum velocity was between 0.78 and 0.88 m/s. The velocity under the effect of the winter monsoon wave was greater than that under the summer monsoon wave.

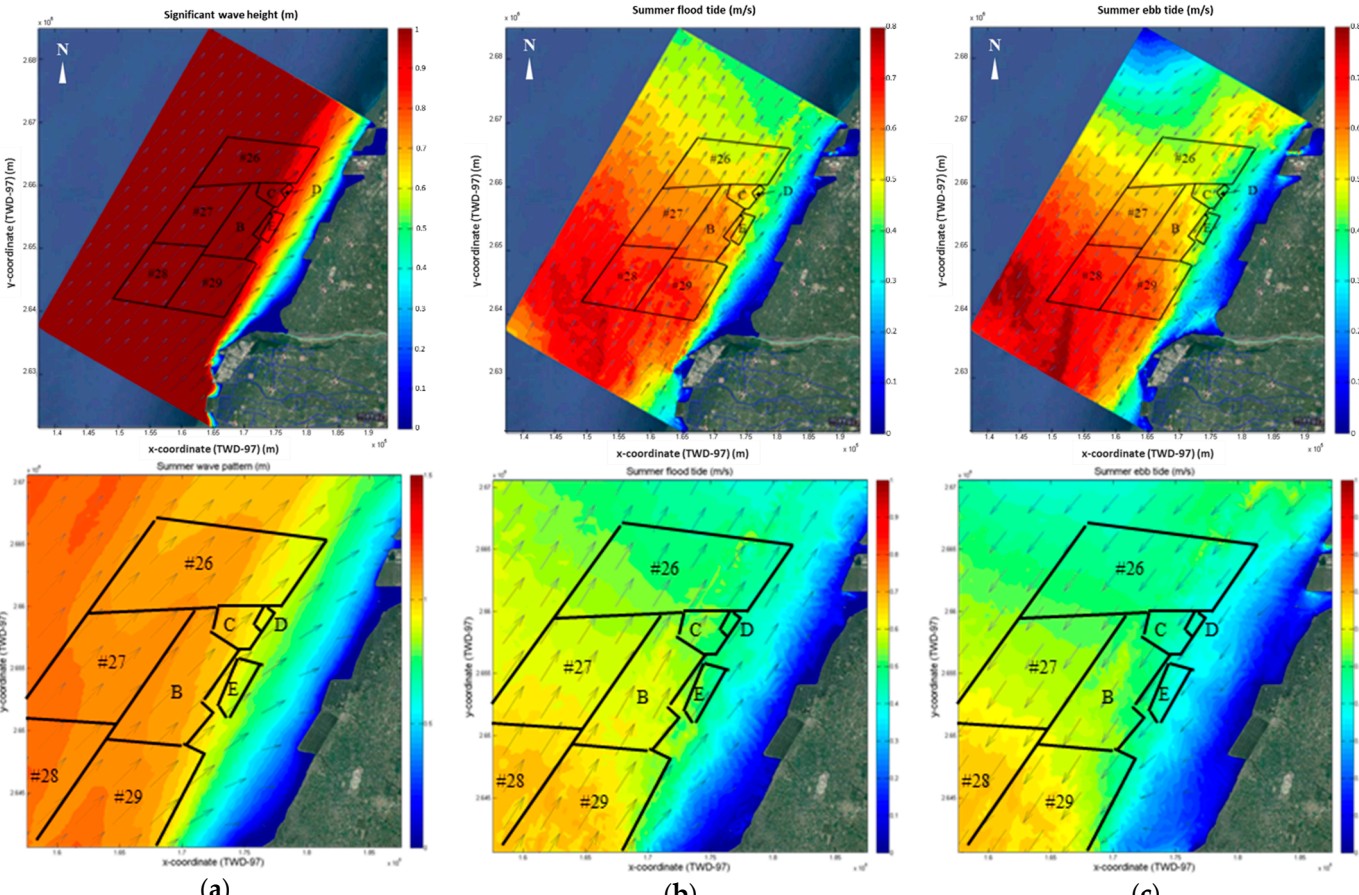

(**a**)　　　　　　　　　　　(**b**)　　　　　　　　　　　(**c**)

**Figure 6.** Simulation results of wave and current under summer conditions. (**a**) Wave height and direction distribution; (**b**) flood tide current pattern; (**c**) ebb tide current pattern.

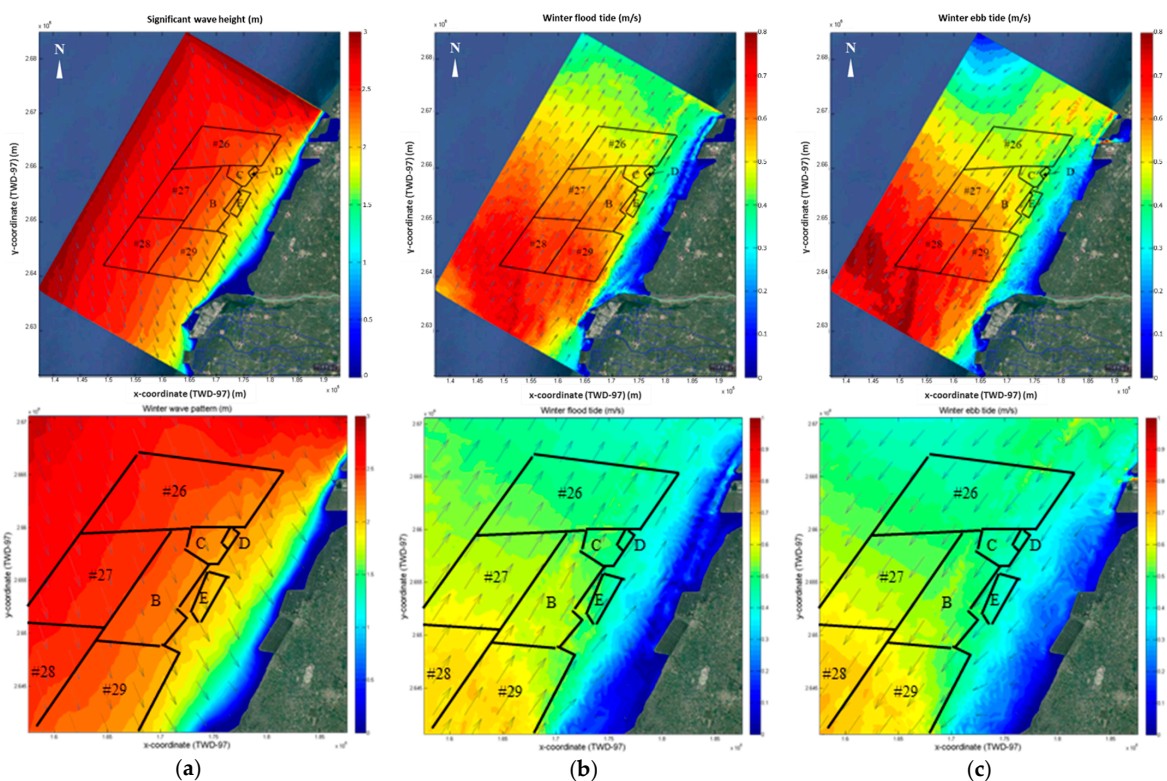

**Figure 7.** Simulation results of wave and current under winter conditions. (**a**) Wave height and direction distribution; (**b**) flood tide current pattern; (**c**) ebb tide current pattern.

## 3. Analysis of the Interaction between the Marine Environment and Wind Turbines

### 3.1. Modeling Procedure and Setting

In this study, seven wind farms close to the coastline were taken as an example (Figure 8). The numerical topography with a total number of 559 wind turbines, as shown in Figure 9, was established for simulation. With reference to the development content of Thousand Wind Turbines Project [32], most offshore wind power developers in Taiwan propose to use the jacket type. This study assumes that the bottom part of the pile foundation is a steel pipe which is placed at the vertex of a 20 m square with a diameter of 3.0 m. After calculating the distribution of the wave pattern, the radiation stress and the water level change mechanism were used to calculate the near-shore currents, and the characteristics of the marine environment change in the sea area after the offshore wind farm is set up are discussed.

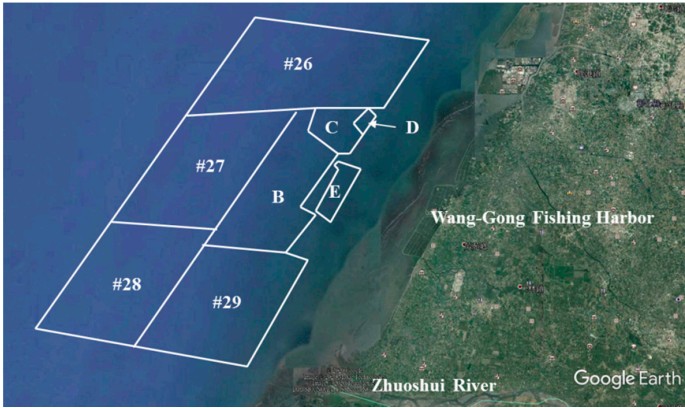

**Figure 8.** The target offshore wind farm in this study. Source: Environmental Impact Assessment Inquiry System (https://eiadoc.eap.gov.tw/EIAWEB/Default.aspx, (accessed on 28 March 2021)).

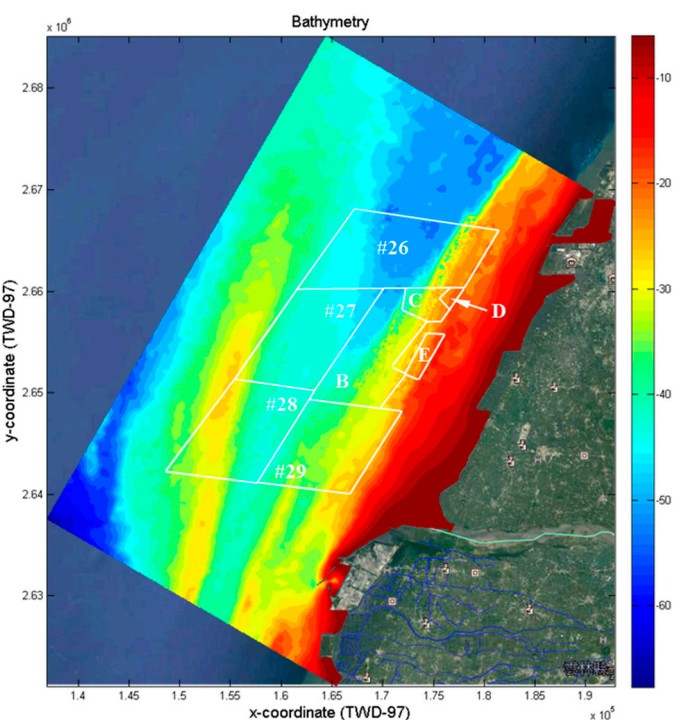

**Figure 9.** Numerical topography after setting 599 wind turbines.

### 3.2. Analysis of the Influence of the Wind Turbine on the Wave and Current Pattern

The wave conditions used in this study are shown in Table 2, as well as the wave theory used to calculate the offshore wavelengths. The calculation result obtained that the wavelength in summer is 65.91 m and the wavelength in winter is 87.75 m, and $\Phi$ is the ratio of the structure diameter to wavelength, which is 1/22 in summer and 1/29 in winter. According to the wave theory, when the diameter of the cylinder in the sea is less than 1/20 of the wavelength, there will be no obvious diffraction and scattering; that is, when the wave passes through the relatively small cylinder, only local reflections and local shadowing effects due to the structure will occur, but its impact range is limited. After the wave passes through the influence range, the local deformation effect is quickly attenuated; that is, its influence is minimal, which is in accordance with the reasonable response of the offshore foundation diameter (3 m) and wavelength ratio between 1/22 and 1/29 in this study. The simulation results after the wind turbines are set up are shown in Figures 10 and 11. Compared with the simulation results of the wave and current patterns without offshore wind turbines, these show that after the establishment of offshore wind turbines, the impact on the wave and current pattern in the sea area is slight and has limited variation characteristics.

**Table 2.** The ratio of offshore wavelength to the basic diameter of the wind turbine.

|  | Period (s) | Wavelength (m) | $\Phi$3.0 m |
|---|---|---|---|
| Summer | 6.5 | 65.91 | 1/22 |
| Winter | 7.5 | 87.75 | 1/29 |

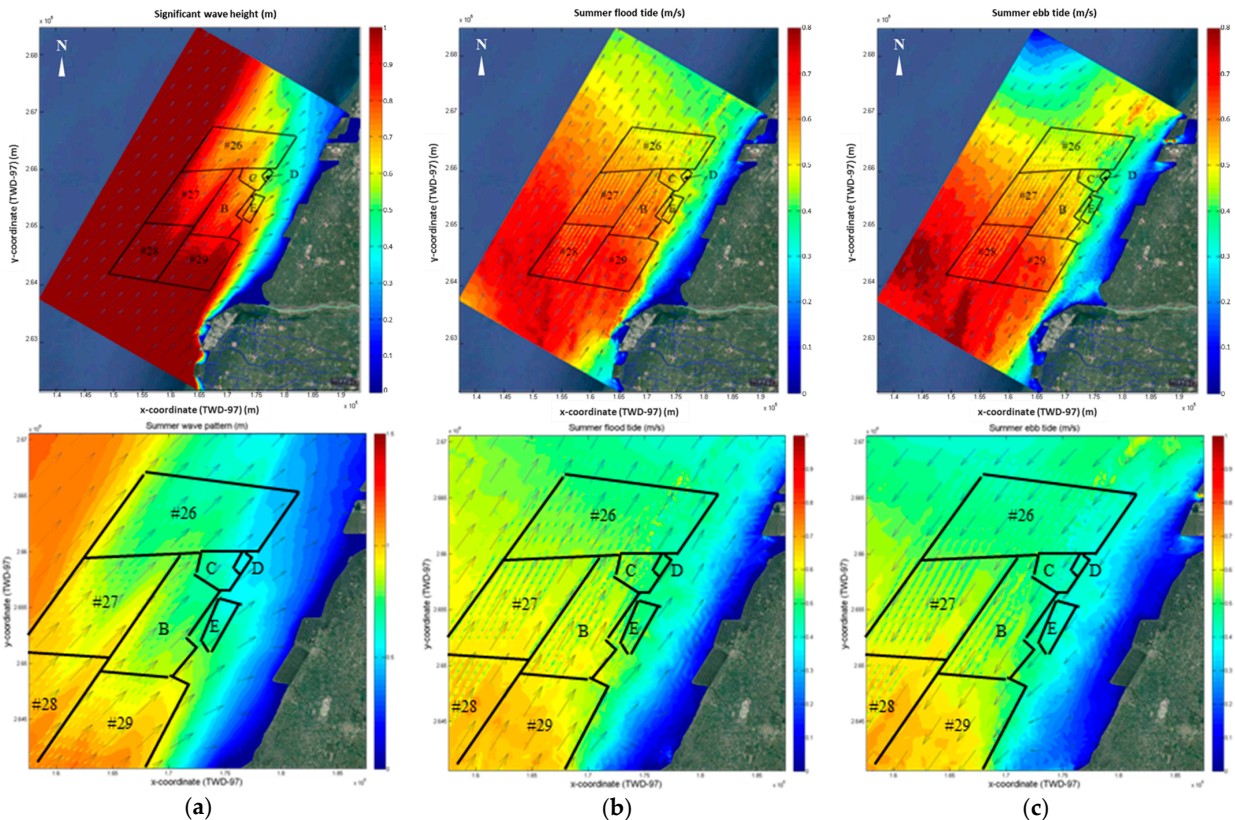

**Figure 10.** Simulation results of wave and current under summer conditions after the establishment of the wind turbine. (**a**) Wave height and direction distribution; (**b**) flood tide current pattern; (**c**) ebb tide current pattern.

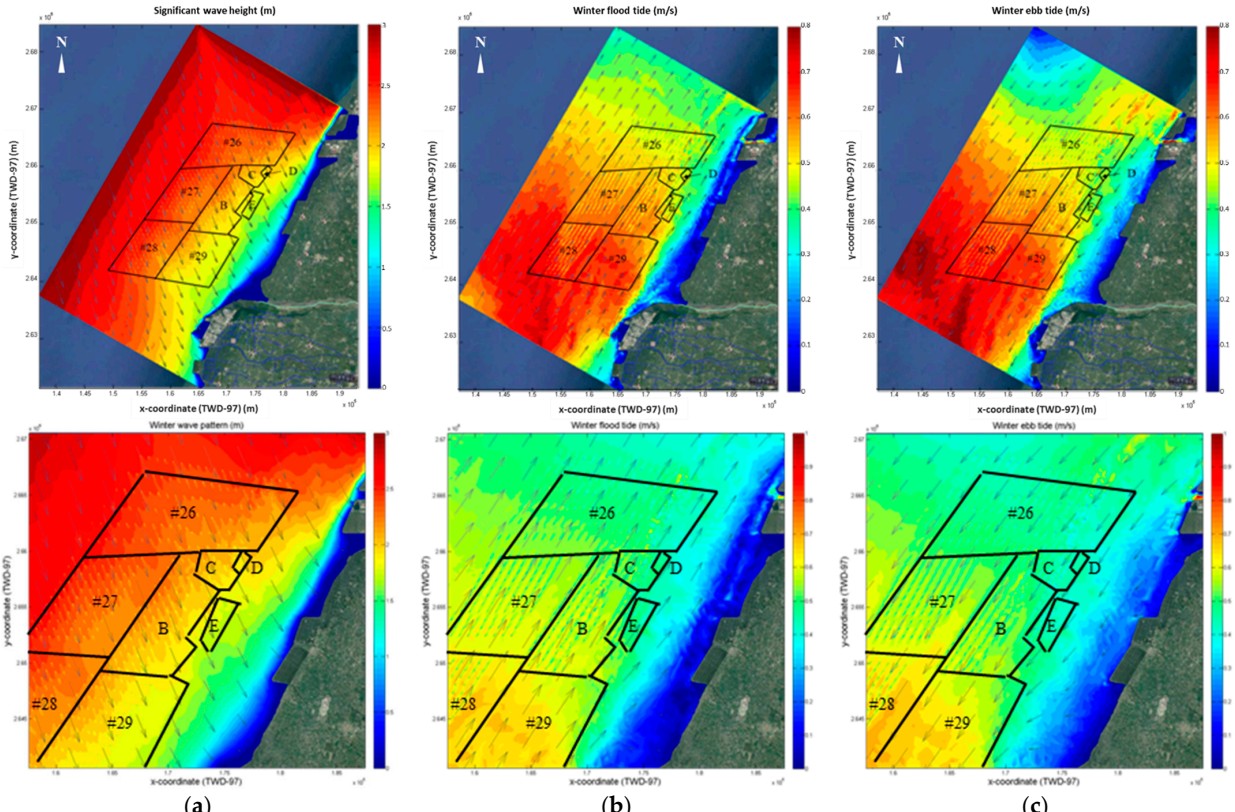

**Figure 11.** Simulation results of wave and current under winter conditions after the establishment of the wind turbine. (**a**) Wave height and direction distribution; (**b**) flood tide current pattern; (**c**) ebb tide current pattern.

### 3.3. Analysis of the Local Wind Turbines Impact on the Marine Environment

In this study, the influence of local wind turbines on the marine environment was taken as an example of the wind farm closest to the shore (code E). Figure 12 is an enlarged view of the simulation results of local wind turbines. The simulation results show that the influence of the wind turbine on the overall wave pattern is only limited to the periphery of the tower, and the change compared to the incident wave height is lowly correlated, which is in line with the description of small diameter structures by wave theory. Since the diameter of the cylinder is smaller than the wavelength, no diffraction occurs behind the cylinder. The influence of the monsoon wave is about four to five times the size of the diameter of the pile (12 to 15 m).

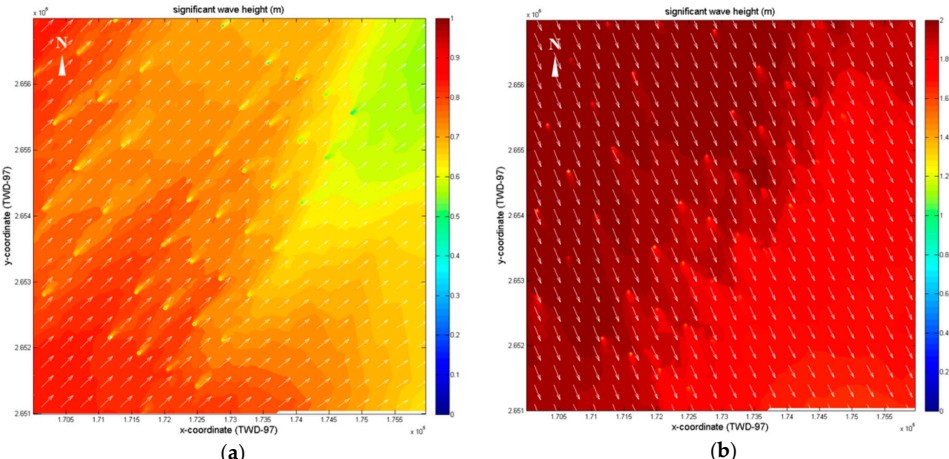

(a)            (b)

**Figure 12.** Simulation results of local wave patterns after wind turbines were established. (**a**) Summer condition; (**b**) winter condition.

Figures 13 and 14 are enlarged views of the simulation results of the current pattern under the local area of the wind turbines. The simulation results show that the simulation results of the current pattern are similar to the wave pattern, and the influence of the wind turbine on the current pattern is limited to the periphery of the tower. After the wind turbine was set up, in the simulated range near the pile, no broken waves occurred, and the influence was about six times the pile diameter (18 m) under the monsoon wave conditions. Overall, the analysis of the simulation results of the wave and current pattern after the offshore wind turbines were established shows that the underwater foundation only affects the local area near the pile structure; that is, there was no obvious change after passing through the limited influence of the underwater foundation.

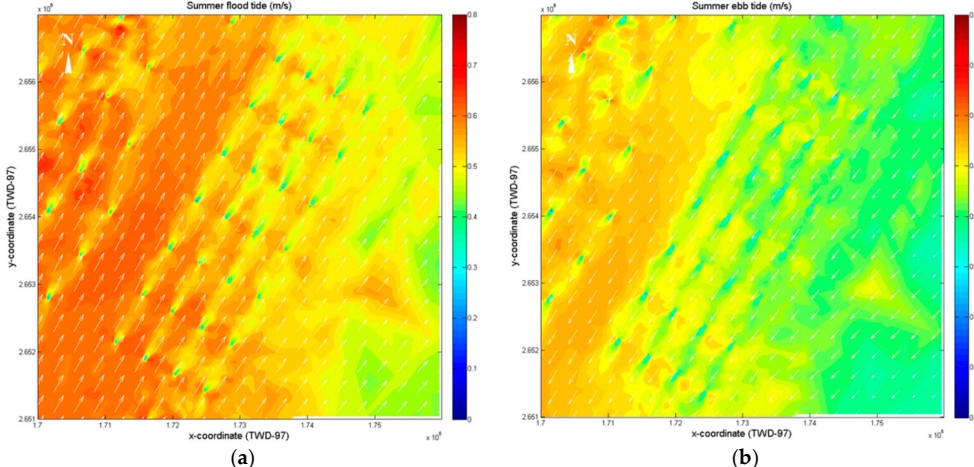

(a)            (b)

**Figure 13.** Simulation results of local current patterns in summer after wind turbines were established. (**a**) Flood tide current pattern; (**b**) ebb tide current pattern.

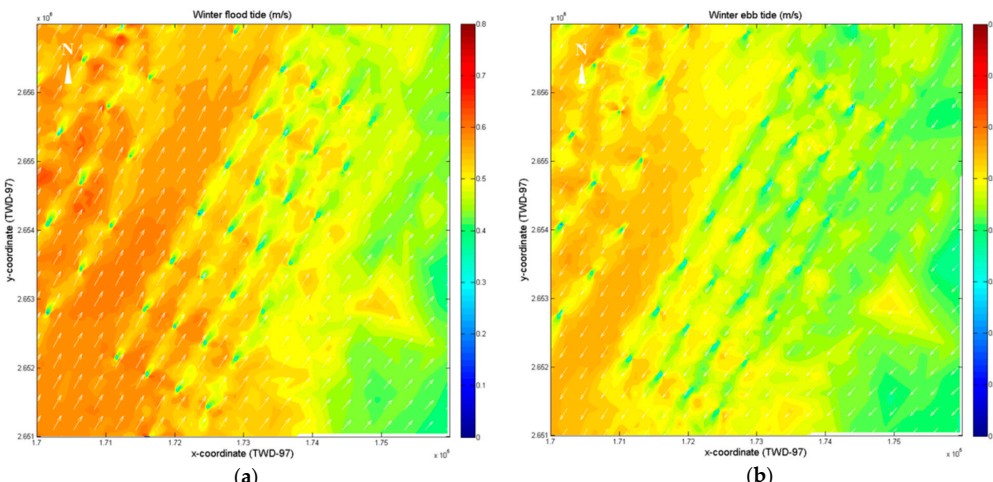

**Figure 14.** Simulation results of local current patterns in winter after wind turbines were established. (**a**) Flood tide current pattern; (**b**) ebb tide current pattern.

*3.4. Discussion*

The analysis of the local simulation results after the offshore wind turbines were established shows that the underwater foundation of the wind turbine only affected the local area near the foundation structure. The overall simulation results show that the waves and currents caused by the wind turbine had a limited influence range, which is probably within the area of the foundation structure of about four to five times the size of its diameter (12–15 m). The influence of the wind turbines' foundation structure on the current was also limited to an area of six times the size of the pile diameter (18 m). These results show that the range of waves and currents affected by the wind turbine occurred in local areas. In order to consider the combination of offshore wind power and traditional fisheries as a form of cooperative development, this study used the case of a circular cage culture that is more common in Taiwan's seas and can effectively resist waves [33]. As shown in Figure 15, this study used the offshore wind field (code E) that is closer to the shore as an analysis case. The arrangement of the wind turbines in the offshore wind farm with code E was such that the parallel shoreline distance was 780 m and the vertical shoreline distance was 480 m. After deducting the range of possible influence of the waves and currents after passing the offshore wind turbine under the influence of monsoon conditions, the estimated range of 349,600 m$^2$ not affected by the offshore wind turbine can be obtained. After referring to the research on the relevant offshore cage culture [34–36], this study took the circular cage culture with a diameter of 20 m as an example and configured it with the cage culture system of Figure 16. From the range of the outer anchoring system, a diameter of approximately 100 m can be obtained. If preparations are made in a regular arrangement, approximately 40 cage cultures can be placed in the area not affected by the wind turbines. If this is expanded to the entire offshore wind farm, it should be possible to configure 720 cage cultures. It is worth mentioning that the use of this configuration does not take into account the future operation of the entire offshore wind farm. If the operation and maintenance of offshore wind farms are not affected, and if the consent of the developer is obtained, it should be possible to use this method to provide economically large-scale farming areas and as a mutually beneficial method for offshore wind power generation and fishery.

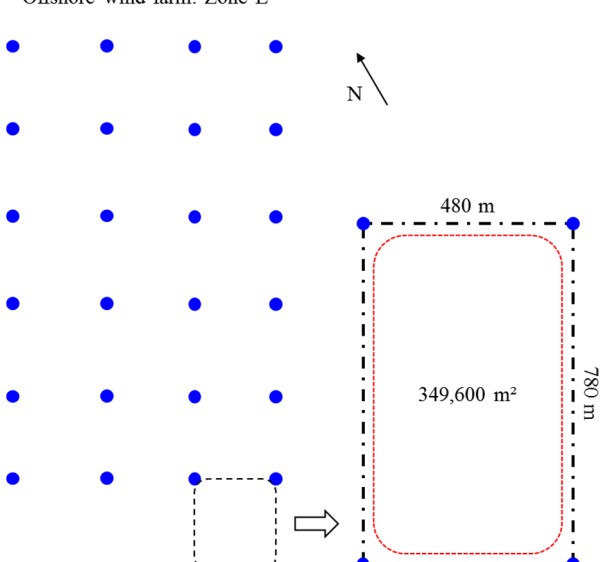

**Figure 15.** Area estimates can be provided within the scope of offshore wind farms.

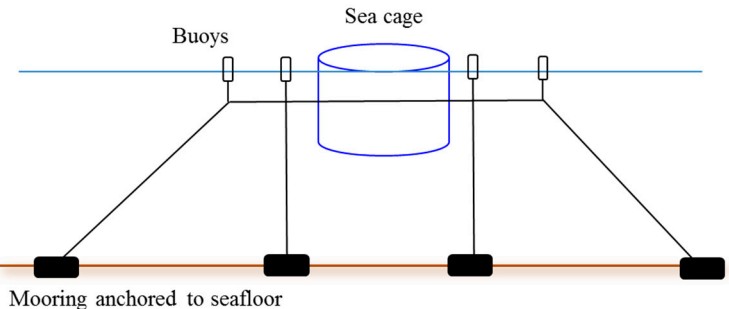

**Figure 16.** Schematic diagram of cage culture system.

## 4. Conclusions

This study used the hydrodynamic model under the influence of wave–current interaction to simulate the wave–current pattern around the structure. The overall simulation results show that the waves and currents caused by the wind turbine had a limited influence range, which was probably within the area of the foundation structure of about four to five times the size of its diameter (12–15 m). The influence of the wind turbine's foundation structure on the current was also limited to an area of six times the size of the pile diameter (18 m). Overall, the analysis of the simulation results of the wave and current pattern after the offshore wind turbines were established shows that the underwater foundation only affected the local area near the pile structure; that is, there was no obvious change after passing through the limited influence of underwater foundation. Since the wind farm (code E) of the research case can be equipped with about 720 cage cultures, if it is extended to other wind farms in the western sea area, it should be able to produce economic-scale farming operations such as offshore wind power and fisheries. However, this study did not consider the future operation of the entire offshore wind farm. If offshore wind farms and fisheries can be mutually beneficial and symbiotic as a development opportunity, it should be considered whether it is possible to carry out offshore cage culture within the offshore wind farm. In the future, it should be possible to conduct follow-up research and analysis without affecting the wind farm maintenance needs and respecting the developer's wishes.

**Author Contributions:** Conceptualization, H.-Y.W. and Y.-C.C.; methodology, Y.-C.C. and H.-M.F.; software, Y.-C.C. and H.-Y.W.; validation, H.-Y.W., H.-M.F. and Y.-C.C.; formal analysis, H.-Y.W.; investigation, H.-Y.W.; resources, Y.-C.C. and H.-M.F.; data curation, H.-Y.W. and Y.-C.C.; writing—

original draft preparation, H.-Y.W. and Y.-C.C.; writing—review and editing, H.-M.F., H.-Y.W. and Y.-C.C.; visualization, Y.-C.C. and H.-M.F.; supervision, Y.-C.C. and H.-M.F.; project administration, Y.-C.C.; funding acquisition, Y.-C.C. All authors have read and agreed to the published version of the manuscript.

**Funding:** This research received no external funding.

**Institutional Review Board Statement:** Not applicable.

**Informed Consent Statement:** Not applicable.

**Data Availability Statement:** Not applicable.

**Acknowledgments:** Thanks to the editor-in-chief and reviewers for their opinions and suggestions on this study, so that the content of this study can be more complete. Thank you also for the contribution of the editorial team for the smooth publication of this paper.

**Conflicts of Interest:** The authors declare no conflict of interest.

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
