# Peer review of "Study on the Coexistence of Offshore Wind Farms and Cage Culture"

_water, doi:10.3390/w13141960_

Round 1

Reviewer 1 Report

The authors used numerical model to simulate the effect of offshore wind farm on the wave and current climates in the vicinity. While this topic is very interesting and has significant impact on the energy industry and the environment, there are many places that need improvement. Please see my comments below:

  1. My biggest concern is how are the wind turbines simulated in the model? What is their impact on the governing continuity and momentum equations? How are they represented in the model? Are they added in as structures? Please clarify.
  2.  Please provide reference to the Jacket type.
  3. What is the total number of grids of the numerical model? How big is the time step? How long did it take to finish the simulation? Please add clarification to the manuscript.
  4. Figure 2: What is the meaning of each color block? Please also highlight the numerical domain in the figure.
  5. Figure 3: Please highlight the location of offshore wind farm.
  6. Line 208 to 209 described the wave information. Is the information of wave at the boundaries? Or of wave at the offshore wind farm? Please add clarification.
  7. Figure 4: Please highlight the location of offshore wind farm.
  8. It is hard to tell the difference between Figure 7 and Figure 3. Please add a map showing the difference between these two figures.
  9. What are the wave lengths at the offshore wind farm in summer and winter?
  10. The cylinder diameter is indeed small compared to the wave length so a single cylinder's effect on the wave field is small. But what about the effect of a group of cylinders?
  11. Comparing Figure 8(a) to Figure 4(a), the difference of the wave fields is big. Please remove the sentences describing how similar these figures are and provide description and explanation of the difference.
  12. Table 1: What is Phi3.0? Please provide definition.

Editorial:

  1. Line 44: coast -> cause.
  2. Line 74-75: Redundant mentioning of "downstream wind turbines".
  3. Eqn. (4), (8), (10), (15), and (16): Please make them align with the rest of the context which are page-centered.
  4. Line 251-255: Repeated sentences.
  5. Line 360: if -> of

Author Response

The authors used numerical model to simulate the effect of offshore wind farm on the wave and current climates in the vicinity. While this topic is very interesting and has significant impact on the energy industry and the environment, there are many places that need improvement. Please see my comments below:

1.My biggest concern is how are the wind turbines simulated in the model? What is their impact on the governing continuity and momentum equations? How are they represented in the model? Are they added in as structures? Please clarify.

  • In the calculation model, the offshore wind turbine is established as a total reflection boundary.

2. Please provide reference to the Jacket type.

  • Added description in line 196-197, and added Figure 3.

3.What is the total number of grids of the numerical model? How big is the time step? How long did it take to finish the simulation? Please add clarification to the manuscript.

  • Modify the content of line 198-203 and add Table 1.

4.Figure 2: What is the meaning of each color block? Please also highlight the numerical domain in the figure.

  • Added instructions at line 183-186.

5.Figure 3: Please highlight the location of offshore wind farm.

  • Increase the area for calculating the wind farm, and it is shown in Figure 4 after modification.

6.Line 208 to 209 described the wave information. Is the information of wave at the boundaries? Or of wave at the offshore wind farm? Please add clarification.

  • The wave condition is entered at the boundary.

7.Figure 4: Please highlight the location of offshore wind farm.

  • 4 and Fig. 5 are modified into Fig. 6&7, and the boundary line of offshore wind farm is added in the range.

8.It is hard to tell the difference between Figure 7 and Figure 3. Please add a map showing the difference between these two figures.

  • The revised figure 7 becomes figure 9.

9.What are the wave lengths at the offshore wind farm in summer and winter?

  • The wavelength in summer is 65.91 m and the wavelength in winter is 87.75 m.

10.The cylinder diameter is indeed small compared to the wave length so a single cylinder's effect on the wave field is small. But what about the effect of a group of cylinders?

  • Since the diameter of the cylinder is small compared to the wavelength, there is no diffraction behind the cylinder. Please find these figures in attached files.

summer

winter

Typhoon

11.Comparing Figure 8(a) to Figure 4(a), the difference of the wave fields is big. Please remove the sentences describing how similar these figures are and provide description and explanation of the difference.

  • In the overall local numerical range, the wave height changes in the monsoon wave is only about 0.01m, while the typhoon wave is about 0.105m. Compared with the incident wave height, the change is not large.

12.Table 1: What is Phi3.0? Please provide definition.

  • Added description at line 281-282.

Editorial:

  1. Line 44: coast -> cause.
  2. Line 74-75: Redundant mentioning of "downstream wind turbines".
  3. Eqn. (4), (8), (10), (15), and (16): Please make them align with the rest of the context which are page-centered.
  4. Line 251-255: Repeated sentences.
  5. Line 360: if -> of

Reviewer 2 Report

In this manuscript, the effects on waves and currents are addressed in the presence of an offshore wind farm in Taiwan. The results of the simulations show that the wave and current fields are not influenced significantly. In addition, the possibility of deploying cages between offshore wind turbines is discussed.

Some recommendations and comments are provided in order to improve the overall quality of the manuscript. In conclusion, the paper is not acceptable in the present form. After corrections, amendments and modifications, including explanations/elaborations of some parts, as listed in the comments below, it could be reconsidered for publication in this journal.

Comments

  1. There are a lot of grammar/expression errors and mistakes in the use of the English language throughout the manuscript making its reading difficult. The whole text should be carefully checked and corrected by a native English speaker.
  2. The title is not representative of the work presented. Emphasis is on the interaction of currents and waves with the offshore wind turbines and not on the coexistence of offshore wind farms and aquaculture. Please consider changing the title or elaborate more the latter aspect.
  3. More details on the model used should be provided. What model was used? In house? Is it a fully coupled system?
  4. What layout was chosen for the offshore wind farm and why?
  5. Provide more details on the wave data used in the scenarios. Why only summer and winter are examined? What is the simulation time for the results presented in Figures 8-12?
  6. References on previous studies related to the impacts of offshore wind farms on waves and currents are rather poor. Consider adding more in the introduction and elaborate what’s new is introduced by the proposed model.
  7. How did the authors conclude that the waves and currents caused by the wind turbine have a limited influence range? It would be helpful if the spatial distribution of flow patterns with and without the wind farm is displayed in two separate figures and in another one show the differences in magnitude and direction.
  8. Are focusing effects and sheltering considered in the wave model?

Author Response

In this manuscript, the effects on waves and currents are addressed in the presence of an offshore wind farm in Taiwan. The results of the simulations show that the wave and current fields are not influenced significantly. In addition, the possibility of deploying cages between offshore wind turbines is discussed.

Some recommendations and comments are provided in order to improve the overall quality of the manuscript. In conclusion, the paper is not acceptable in the present form. After corrections, amendments and modifications, including explanations/elaborations of some parts, as listed in the comments below, it could be reconsidered for publication in this journal.

Comments

1.There are a lot of grammar/expression errors and mistakes in the use of the English language throughout the manuscript making its reading difficult. The whole text should be carefully checked and corrected by a native English speaker.

  • English edited through MDPI.

2.The title is not representative of the work presented. Emphasis is on the interaction of currents and waves with the offshore wind turbines and not on the coexistence of offshore wind farms and aquaculture. Please consider changing the title or elaborate more the latter aspect.

  • With the future development of large-scale wind turbines, the offshore wind turbines planned by Taiwan's offshore wind power developers re separated by about 1 km. This important area for the development of offshore wind power in the future is also one of the most prosperous areas for offshore fishing operations in western Taiwan. In highly overlapping economic sea areas, the development of offshore wind power may impact the fishing and ecological environment in the same sea area. The development utilized by the comprehensive planning of ocean space was thought to be in the long-term public interest for the use of the sea area of the offshore wind power. So we consider if it is possible to establish a cage culture system to assist fishermen to transform their livelihoods into sustainable fisheries and to transform offshore wind power and traditional fisheries into cooperative developments.

3.More details on the model used should be provided. What model was used? In house? Is it a fully coupled system?

  • Research team develops model based on RCPWAVE.

4.What layout was chosen for the offshore wind farm and why?

  • Focusing on the areas announced by Taiwan, and selecting areas with intensive fishing activities for research, it is hoped that the losses caused by wind turbine construction to fishing activities can be compensated.

5.Provide more details on the wave data used in the scenarios. Why only summer and winter are examined? What is the simulation time for the results presented in Figures 8-12?

  • The typhoon is a short-term effect. During the typhoon, the cage may be closed up or shallowed into the seabed. The longer-term influence of Taiwan is caused by the monsoon, and the wind direction of Taiwan in summer is completely different from that in winter, so only the result of the monsoon is considered.

6.References on previous studies related to the impacts of offshore wind farms on waves and currents are rather poor. Consider adding more in the introduction and elaborate what’s new is introduced by the proposed model.

  • This study used a hydrodynamic model that included wave and current characteristics for calculation. The numerical model analyzed the impact of offshore wind turbines on the wave–current pattern.

7.How did the authors conclude that the waves and currents caused by the wind turbine have a limited influence range? It would be helpful if the spatial distribution of flow patterns with and without the wind farm is displayed in two separate figures and in another one show the differences in magnitude and direction.

  • Because the diameter of the cylinder is small compared to the wavelength, there is no diffraction behind the cylinder. The influence range of the monsoon wave is about 4 to 5 cylinder diameters, while the influence of the typhoon wave is five cylinder diameters. Please find these figures in the attached file.

summer

winter

Typhoon

8.Are focusing effects and sheltering considered in the wave model?

  • In the calculation model, the offshore wind turbine is established as a total reflection boundary.

Reviewer 3 Report

The paper considers the water waves and current at the site where fish cages are commissioned. The bottom bathymetry is taken into consideration via the mild slope equations. In general, the study is valuable, and fits into the scope of the journal. The only issue is that the title is misleading. In the paper, authors only consider the environments of the fish cage, whereas the corresponding loading and response are missing. The title should reflect this.

Author Response

The paper considers the water waves and current at the site where fish cages are commissioned. The bottom bathymetry is taken into consideration via the mild slope equations. In general, the study is valuable, and fits into the scope of the journal. The only issue is that the title is misleading. In the paper, authors only consider the environments of the fish cage, whereas the corresponding loading and response are missing. The title should reflect this.

  • With the future development of large-scale wind turbines, the offshore wind turbines planned by Taiwan's offshore wind power developers re separated by about 1 km. This important area for the development of offshore wind power in the future is also one of the most prosperous areas for offshore fishing operations in western Taiwan. In highly overlapping economic sea areas, the development of offshore wind power may impact the fishing and ecological environment in the same sea area. The development utilized by the comprehensive planning of ocean space was thought to be in the long-term public interest for the use of the sea area of the offshore wind power. So we consider if it is possible to establish a cage culture system to assist fishermen to transform their livelihoods into sustainable fisheries and to transform offshore wind power and traditional fisheries into cooperative developments.

Round 2

Reviewer 1 Report

The authors have responded to my comments. Some are well addressed, while some do not answer my questions. I have the following comments upon my 2nd review:

  1. What is the location of the measurement shown in Figure 5?
  2. How long did the simulation take to run?
  3. Line 283-284: Please provide the reference of phi values.
  4. Line 309: "The simulation results show that the influence of the wind turbine on the overall wave pattern is only limited to the periphery of the tower". However, comparing the plots of significant wave height in Figure 6(a) and Figure 10(a), the change is big and extends to the northeast boundary of the domain. Please acknowledge it and add clarification. Please consider adding a map showing the difference of significant wave height distribution under the with and without wind farms conditions. Without certain clarification, the conclusion that the effect of offshore wind farms is limited to local area is baseless. 

Editorial: 

  1. The superscripts on U, x, and y in Section 2.1 are not readable.
  2. Figure 4: Please add a legend for the color bar and units for the x- and y- axes.
  3. Figure 5: ADCP cannot measure water level. So please change the legend of measurement data in the first subplot to "Measurement" or something similar.

Author Response

The authors have responded to my comments. Some are well addressed, while some do not answer my questions. I have the following comments upon my 2nd review:

  1. What is the location of the measurement shown in Figure 5?

The measurement position of ADCP is in #26 area, the coordinates are 176881, 2661049 (TWD-97)

  1. How long did the simulation take to run?

A case is about 4~5 days.

  1. Line 283-284: Please provide the reference of phi values.

Φ is the ratio of the structure diameter to wavelength, , is the structure diameter divided by wavelength.

  1. Line 309: "The simulation results show that the influence of the wind turbine on the overall wave pattern is only limited to the periphery of the tower". However, comparing the plots of significant wave height in Figure 6(a) and Figure 10(a), the change is big and extends to the northeast boundary of the domain. Please acknowledge it and add clarification. Please consider adding a map showing the difference of significant wave height distribution under the with and without wind farms conditions. Without certain clarification, the conclusion that the effect of offshore wind farms is limited to local area is baseless. 

Editorial: 

  1. The superscripts on U, x, and y in Section 2.1 are not readable.

Check.

  1. Figure 4: Please add a legend for the color bar and units for the x- and y- axes.

Modify.

  1. Figure 5: ADCP cannot measure water level. So please change the legend of measurement data in the first subplot to "Measurement" or something similar.

The ADCP used has a pressure sensor that can measure the water level.

Reviewer 2 Report

The overall quality of the revised manuscript was improved. Although some comments were not sufficiently addressed by the authors (e.g. comments No 3, 4, 6, 8) according to the opinion of this reviewer, the manuscript is accepted in its present form.

Author Response

We appreciate for Reviewers’ warm work earnestly, and comments No 3, 4, 6, 8 will served as our future studies. Once again, thank you very much for your comments and suggestions.
